# Recovery Nystagmus in Vestibular Neuritis with Minimal Canal Paresis. Clinical Observation and Interpretation

**DOI:** 10.3390/brainsci12010110

**Published:** 2022-01-14

**Authors:** Eleni Zoe Gkoritsa

**Affiliations:** Independent Researcher, 21 Petrou Mpoua Street, 22131 Tripoli, Greece; zoilen@hotmail.com

**Keywords:** recovery nystagmus, vestibular neuritis, minimal canal paresis

## Abstract

Recovery nystagmus in vestibular neuritis patients is a reversal of spontaneous nystagmus direction, beating towards the affected ear, observed along the time course of central compensation. It is rarely registered due either to its rarity as a phenomenon per se, or to the fact that it is missed between follow-up appointments. The aim of the manuscript is to describe in detail a case of recovery nystagmus found in an atypical case of vestibular neuritis and discuss pathophysiology and clinical considerations regarding this rare finding. A 26-year-old man was referred to our Otorhinolaryngology practice reporting “dizziness” sensation and nausea in the last 48 h. Clinical examination revealed left beating spontaneous nystagmus (average slow phase velocity aSPV 8.1°/s) with absence of fixation. The head impulse test (H.I.T.) was negative. Cervical vestibular evoked myogenic potentials (cVEMP) and Playtone audiometry (PTA) were normal. Romberg and Unterberger tests were not severely affected. A strong directional preponderance to the left was found in caloric vestibular test with minimal canal paresis (CP 13%) on the right. The first follow-up consultation took place on the 9th day after the onset of symptoms. Right beating weak (aSPV 2.4°/s) spontaneous nystagmus was observed with absence of fixation, whereas a strong right directional preponderance (DP) was found in caloric vestibular test. A brain MRI scan was ordered to exclude central causes of vertigo, which was normal. The patient was seen again completely free of symptoms 45 days later. He reported feeling dizzy during dynamic movements of the head and trunk for another 15 days after his second consultation. The unexpected observation of nystagmus direction reversal seven days after the first consultation is a typical sign of recovery nystagmus. Recovery nystagmus (RN) is centrally mediated and when found, it should always be carefully assessed in combination with the particularities of vestibular neuritis.

## 1. Introduction

Recovery nystagmus (RN) has been described by Stenger in 1959 [1] with the term “erholungs Nystagmus” which is still used by neurootologists worldwide. In 2019, the Bárány Society [2] included recovery nystagmus in the classification and definitions for nystagmus and nystagmus-like movements. It is defined as “spontaneous peripheral vestibular nystagmus that has reversed direction after a period of time (usually hours or days, depending on the cause) and is attributed to recovery from an underlying vestibular disorder causing an initial inhibitory nystagmus”.

The characteristics of spontaneous peripheral vestibular nystagmus of inhibitory type (previously called “paretic”) occurring after an acute unilateral vestibular loss are well defined. The nystagmus is jerk type, i.e., consists of a fast and a slow velocity phase. Its direction is mostly horizontal (with a torsional component if all three semicircular canals are affected) and it obeys Alexander’s law: (a) the fast component is beating away from the lesioned side, (b) nystagmus intensity increases when looking in the fast phase direction and decreases when looking in the slow phase direction, (c) nystagmus intensity increases with removal of fixation. By convention, the direction of nystagmus is the direction of its fast component. Recovery nystagmus direction is towards the lesioned ear.

RN results from the persistence of a degree of central compensation for an initial imbalance in vestibular tone after the need for this amount of compensation is lessened or absent. Recovery nystagmus primarily occurs when the initial disease phase is sustained, often for hours or days, and the recovery is rapid (minutes to hours, perhaps days) [2].

RN is not frequently registered for two reasons: (A) it does not take place in every individual with acute vestibular deafferentation (AVD) [3,4]. (B) There is a wide variability regarding the time of its appearance and the duration of its presence. Lee et al. [5] found a range of recovery nystagmus detection time at 17.56 ± 14.16 days after the onset of AVD. Lange [6] reports registration of recovery nystagmus from as early as 5 days to very late after injury (2 years). Consequently, there is a high probability that it is missed between follow-up appointments.

This study aims to present a patient in whom recovery nystagmus was recorded as an unexpected finding on the second consultation seven days after the initial one. The early pathogenetic mechanisms of central compensation are discussed together with the pathophysiology of recovery nystagmus.

## 2. Case Presentation

A 26-year-old man was referred to our Otorhinolaryngology practice reporting “dizziness” sensation and nausea with two vomiting episodes in the last 48 h. The onset was abrupt with sudden vomiting and nausea deteriorating with head motion. By the time he was medically assessed (48 h later) the symptoms had slightly improved, but he reported persistence of nausea. This was the first experienced vertiginous episode, and the patient had no history of motion sickness or any other vestibular related clinical entities. Clinical examination revealed left beating spontaneous nystagmus (average slow phase velocity (aSPV) 8.1°/s) with absence of fixation (Figure 1A), while there was no clinically observed spontaneous nystagmus with fixation. Left beating horizontal-torsional spontaneous nystagmus was observed again at the right posterior position of the Dix-Hallpike test with no fatigue or latency. The head impulse test (HIT) was negative. The skew deviation test and quantitative vestibular ocular tests (saccades, smooth pursuit, optokinetic nystagmus suppression) were unremarkable. Cervical vestibular evoked myogenic potentials (cVEMP) (Interacoustics Eclipse EP/25) showed an asymmetry Ratio of 20% (non-significant) and Playtone audiometry (PTA) was normal (i.e., average 500 Hz, 1000 Hz, 2000 Hz and 4000 Hz threshold < 20 dB HL). The direction of left beating nystagmus was preserved after provocation by head shake. A sensation of imbalance was reported on the Romberg test, but no body sway was observed. A 14° rotation to the right with 10 cm of displacement was observed on Unterberger test (20 steps). A strong left directional preponderance (80%) was found in bithermal caloric vestibular test with a concomitant right minimal canal paresis (13%). Air caloric irrigations were performed (46° Celsius -warm and 28° Celsius -cold) (Difra Coolstar calorics irrigator). The duration ofeach irrigation was 40 s and nystagmic reactions were registered with videonystagmography (Interacoustics Eclipse EP/25). The cerebellar function tests (for elimination of dysdiadochokinesia and ataxia) finger counting and) were normal. He was treated with dimenydrinate initially (120 mg bd) but after 48 h due to complaints of constipation, his treatment switched to beta-histine (16 mg bd) for another 4 days. The first follow-up consultation took place on the seventh day (9th day after the onset of symptoms). Nausea and vomiting had stopped but he still felt dizzy when moving his head or bending over. Right beating weak (aSPV 2.4°/s) spontaneous nystagmus was observed with absence of fixation (Figure 1B), whereas a strong right directional preponderance was found in caloric vestibular test (DP 45%). Clinical evaluation of vestibular function and balance was normal apart from a slight deviation to the left on Fukuda stepping test (10°). To exclude central causes of vertigo, a brain MRI scan was ordered, which was normal. He was instructed to perform simple Cawthorn–Cooksey exercises [7] at home for about ten days. The patient was seen again completely free of symptoms 45 days later. He reported feeling dizzy during dynamic movements of the head and trunk for another 15 days after his second consultation. The test battery mentioned above was repeated and no vestibular or balance abnormality was found. Recorded spontaneous nystagmus and caloric responses of the patient are reported on Table 1.

## 3. Discussion

Vestibular neuritis is an acute vestibular syndrome (AVS) with vertigo, nystagmus, gait disturbance and autonomic symptoms (nausea and vomiting), all of which must by convention last more than 24 h [8,9]. The most valid etiopathogenesis theory is the reactivation of a latent herpes simplex 1(HSV-1) inhabiting the vestibular ganglion. Superior branch involvement (affecting the horizontal semicircular canal and/or anterior semicircular canal and/or utricle) is the most frequent variant 57% [10]. Video head impulse test studies in humans [11] showed that the horizontal semicircular canal (SCC) is affected more frequently than the anterior SCC whereas deficits of the utricle are less common. AVS involving both divisions of the vestibular nerve constitute the second most frequent pattern of vestibular neuritis (28.5%). Complete inferior division involvement is the scarcest (5.7%). On the other hand, the severity of anatomical damage of the nerve varies. Histological studies focusing on the anatomical details of the branches of the vestibular nerves and vasculature of the labyrinth suggest that in mild to moderate cases of vestibular neuritis, a more distal involvement of the nerve could lead to incomplete and selective distal branch lesions [12]. From another angle, the distribution of afferent fibers deriving from each vestibular organelle is decisive for the sensitivity of the latter to damage. The more broadly distributed (in a spatial sense) the afferent fibers from an organelle are, the less likely it is that the entire pathway will be damaged by acute vestibular neuritis [13].

Caloric testing has been for many decades the main laboratory test for assessment of the vestibulo-ocular reflex (VOR) function of horizontal SCC. Arbitrary cut off points have been defined to decide canal paresis. Strupp et al. [14] report a value of >25% of asymmetry between the two sides. This value can vary slightly according to the normative data of each laboratory.

The criteria for the diagnosis of Vestibular Neuritis were introduced in 1969 by Coats [15]. Kim et al. [16] report that in practice there are no strict criteria in the diagnosis of this clinical entity. According to Cooper [17] the sudden onset of vertigo and the absence of auditory and neurological symptoms and signs are the most important. Given the knowledge that Acute Vestibular Syndrome (AVS) can affect partially the vestibular nerve and organs, minimal canal paresis can be interpreted as a partial and rapidly recovering damage in some fibers of the superior vestibular nerve.

This is an interesting case of a young man with a first episode of vestibular neuritis symptomatology. The particularities of his clinical findings are suitable for scientific reasoning trying to figure out the underlying pathophysiology. The two main points of interest are the minimal canal paresis (13%) with concomitant directional preponderance (DP) and spontaneous nystagmus (SN) at first consultation and the unexpected finding of inversion of SN direction seven days later. It is well established that a negative head impulse test leads to suspicion of a central cause. Nevertheless in case of partial vestibular deficits the sensitivity of bedside HIT (without video registration) is considerably lower because the residual peripheral function results in a smaller gain asymmetry of the VOR [18]. A negative bedside HIT is therefore nor surprising in the present case where peripheral deficit at the time of examination is below the numerical limits of clinical significance. On the other hand, the option of an MRI was reasonable in the presence and inversion of spontaneous nystagmus together with a negative HIT. Normal cVEMP is an indication of a normally functioning inferior vestibular nerve which allows all clinical assumptions to focus on the superior division of the vestibular nerve.

Differential diagnosis had to exclude other possible causes of acute vestibular insult such as migrainous vertigo, and Menière’s disease. Medical history was negative regarding migraine, there was a lack of symptoms from the anterior labyrinth (i.e., tinnitus and hearing loss) and his vertiginous sensations lasted for twenty days, which makes the diagnosis of migrainous vertigo and hydrops less probable. The patient’s reported dizziness for 48 h which finally turned out to cause symptoms for a total of twenty days does not comply either with Diagnostic Criteria for Vestibular Migraine as defined by the Bárány Society [19], or with Diagnostic Criteria for Menière’s disease [20]. In case of the latter, episodic vertigo may precede the onset of hearing loss by several weeks or months, but tinnitus or aural fullness is usually associated with the first episode of vertigo. Additionally, the negative history of motion sickness together with the fact that head shake nystagmus was absent in the last consultation (50 days later), argue against the possibility of recurrent spontaneous nystagmus with interictal head shake nystagmus, a clinical entity described by Lee et al. [21].

The most probable clinical assumption, therefore, is an episodic insult of the vestibular end organ which recovered to almost complete function within 48 h.

The mechanism of spontaneous nystagmus generation as a result of peripheral vestibular deafferentation has been reported with precision [22] (Figure 2). Spontaneous nystagmus is the most characteristic of the static signs of acute vestibular deafferentation. Static signs are observed in the absence of head movement, affecting head and body posture and eye movement (spontaneous nystagmus, ocular cyclotorsion, postural instability).

The commissural inhibitory connections between the two medial vestibular nuclei play a major role in the beginning of vestibular compensation. As a matter of fact, compensation aims at the restoration of resting firing discharge rate of Type I cells of the ipsilesional MVN and further rebalance between the two medial vestibular nuclei. Reaction mechanisms of vestibular compensation can be distinguished into immediate (appearing in the first hours or days after deafferentation) and delayed late ones (appearing in the following weeks to months) [23]. The most immediate mechanisms that lead to reappearance of discharge activity of the ipsilesional nucleus are an increase in the intrinsic electrophysiological excitability of MVN Type 1 cells and the rapid downregulation of the functional efficacy of receptors (GABA-ergic and glycinergic) on the ipsilesional MVN inhibitory Type II neuron cells by commissural homeostatic mechanisms.

In the case at issue, recovery begun apparently earlier than 48 h (having as a result the restoration of the function of the peripheral organ, as is shown by minimal CP) and evolved in the week that followed, having as a result not just the disappearance of SN but also the appearance of contraversive recovery nystagmus. Kim et al. [16] who have studied a group of patients with minimal CP vestibular neuritis report the occurrence of spontaneous nystagmus and contralateral together with the finding of minimal CP. They report the observation of recovery nystagmus in these patients in a statistically higher grade than the group of patients with CP > 25% (the lower limits of their lab). Lee et al. [5] reported no change of symptoms at the time of recovery nystagmus in their patients’ group, as has happened with our patient.

Recovery of peripheral vestibular function is among a possible clinical outcome of AVS. The restoration of partial end organ damage by spontaneous synaptic repair in the sensory epithelia has been confirmed [24]. Experimental studies with excitoxically caused lesions of the vestibular afferent nerve fiber terminals showed the propensity of vestibular hair cells and primary neurons to restore their synaptic contacts following selective damage. In what concerns compensation mechanisms, in addition to events occurring in the brainstem, the role of afferent stimuli from the remaining or newly restored functional peripheral synapses in the restoration of electrophysiological equilibrium between the vestibular nuclei is probably also of high importance.

Tighilet et al. [24] claim that the interference of peripheral signaling into vestibular nuclear complex while the latter is in the process of vestibular compensation, can modulate the process of compensation itself. Peripheral functional recovery in the first hours or days following the initial insult can perturb and possibly delay the establishment of compensation mechanisms in the Vestibular Nuclei. This is the most accurate assumption which could justify the co-existence of minimal CP and spontaneous nystagmus at the absence of visual fixation (i.e., although peripheral organ function has almost recovered the recovery of Vestibular Nucleus resting activity is on the way, but has not yet reached such a level of intensity as to annul SN).

Seven days later, the occurrence of recovery nystagmus, and the consequent DP on the ipsilesional side, is a sign of overstimulation of the ipsilesional Vestibular Nucleus Type I neurons. This can be explained by the finalization of the restoration of the peripheral lesion, boosting the ipsilateral MVN with peripheral input (Figure 3).

Another factor which could be interfering with the firing rates of vestibular nuclei is beta-histine. Beta-histine is both a partial histamine agonist and a H3 receptor antagonist. H3 receptors are found presynaptically in the histaminergic ends of the neurons of the tuberomammillary nuclei in the thalamus and as heteroreceptors in non histaminergic nerve terminals (GABAergic nerve fibers of the commissural system) of the vestibular nuclei (Figure 2). The major central target of beta-histine treatment is the pre-synaptic histamine H3 autoreceptor. The drug-induced blockade of the latter produces an up-regulation of the histamine neuron activity, thus increasing histamine turnover and release of histamine in the VN (excitatory effect on the Type I cells), which in turn inhibits the release of GABA via the H3 heteroreceptors reducing the activity of inhibitory Type II cells (disinhibition of the Type I cells) [25]. The histamine induced modulation of GABA release after acute unilateral vestibular loss could be a very early mechanism of vestibular compensation, as mentioned also earlier. In the present case, beta-histine could contribute to the readiness of medial vestibular nucleus to produce recovery nystagmus via the increased discharging activity of the disinhibited Type I cells.

Experimental data in humans measured the evolution of static signs of AVS such as subjective visual vertical, SN and ocular cyclotorsion and found the beginning of improvement as early as 4 days after the beginning of treatment [25]. The dosage used (48 mg tid) was much higher than the dosage of our patient (16 mg bd) but given the subtle input balances in the minimal CP vestibular neuritis, the possibility of efficacy of beta-histine cannot be excluded.

## 4. Conclusions

Recovery nystagmus in a patient with minimal canal paresis acute vestibular syndrome is of particular interest in the details of its pathophysiologic mechanisms. The case at issue complies with clinical and laboratory observations of previous researchers. Given the existing knowledge about the early mechanisms of central vestibular compensation, recovery nystagmus generation is better understood. A perplexing clinical finding at first, it can boost the certainty of diagnosis together with the rest of vestibular test battery and close clinical observation.

## Figures and Tables

**Figure 1 brainsci-12-00110-f001:**
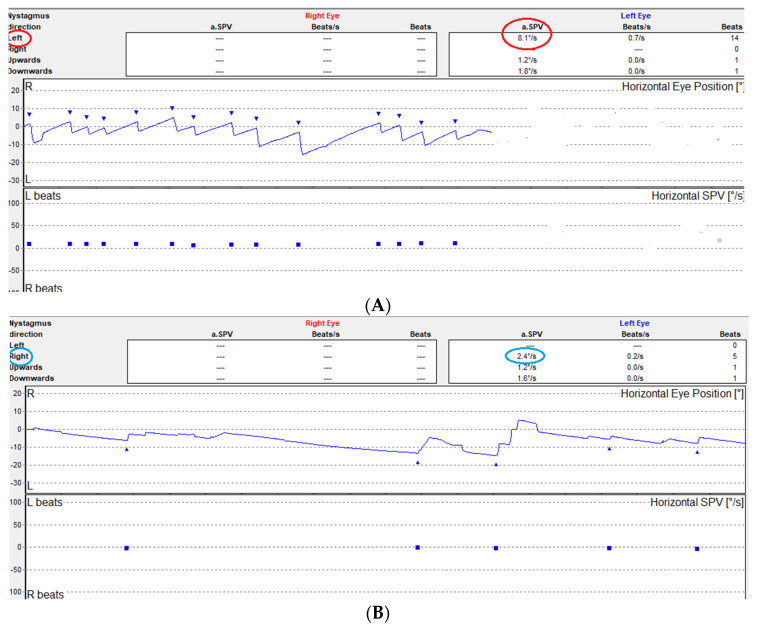
(**A**) Spontaneous nystagmus recorded with absence of fixation 48 h after the vestibular insult (first consultation). (**B**) Recovery spontaneous nystagmus recorded with absence of fixation seven days later (second consultation).

**Figure 2 brainsci-12-00110-f002:**
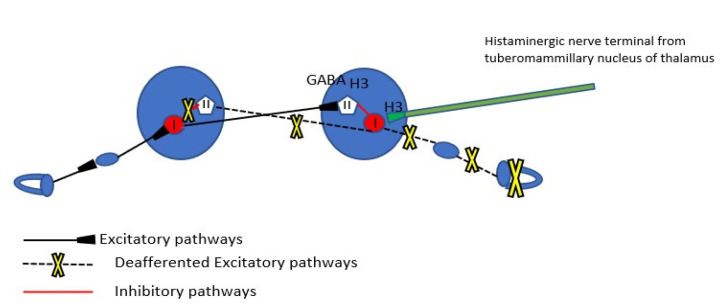
The commissural pathway modulations immediately after unilateral vestibular deafferentation. Type I neurons in the MVN of the ipsilesional side are silenced because of interruption of the peripheral vestibular input, but also by the inhibitory activity of Type II neurons of the same nucleus which are not only spared but potentiated by the exacerbated activity of contralateral Type I neurons which in turn are disinhibited by the disfacilitation of Type II inhibitory neurons of the healthy Medial Vestibular Nucleus (MVN). GABAergic receptors and H3 heteroreceptors (presynaptic) from commissural fibers to Type II neurons are also noted. The precynaptic H3 receptors on the histaminergic nerve terminals from the tuberomamilary nucleus of thalamus are noted (see below).

**Figure 3 brainsci-12-00110-f003:**
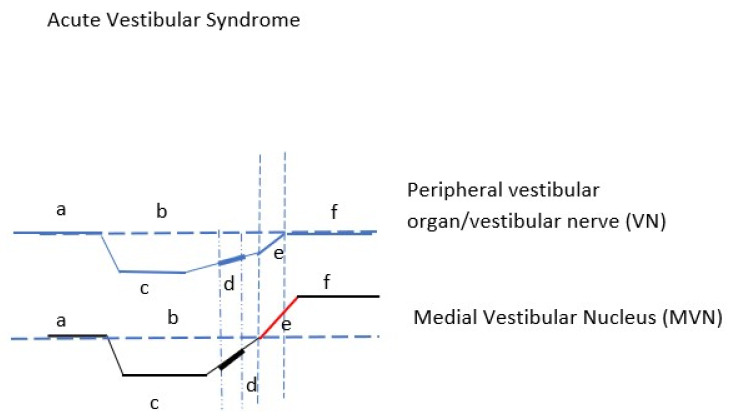
Schematic representation of vestibular nerve (VN) and medial vestibular nucleus (MVN) in acute vestibular syndrome. VN (a) peripheral organ normal function (b) activity line at rest, (c) peripheral organ deafferentation/hypofunction, (d) end organ begins to restore function, (e) final recovery of function, (f) peripheral organ function restored. MVN (a) nucleus baseline activity, (b) baseline activity line, (c) MVN discharge rate ceases, (d) beginning of resting discharge activity—beginning of central compensation, (e) the restoration of end organ function overstimulates the nucleus, (f) recovery nystagmus phase-nucleus hyperfunction. The thick sections in the restoration, (d) curves of VN and MVN represent the first consultation time point of the patient. Peripheral organ has regained much of its function, whereas the ipsilateral MVN begins to regain its intrinsic activity, not reaching yet the point of annulation of spontaneous nystagmus.

**Table 1 brainsci-12-00110-t001:** Spontaneous nystagmus and caloric responses in the first and follow-up consultations of the patient.

Time from Vestibular Insult	SN (aSPV)	CP	DP
48 h	8.1°/s Left	13% Right	80% Left
9 days	2.4°/s Right	2% Right	45% Right
50 days	-	0%	8% Right

SN: spontaneous nystagmus, aSPV: average slow phase velocity, CP: canal paresis, DP: directional preponderance.

## Data Availability

Data are available if requested for at the e-mail of corresponding author: zoilen@hotmail.com.

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
