# Peer review of "Recovery Nystagmus in Vestibular Neuritis with Minimal Canal Paresis. Clinical Observation and Interpretation"

_brainsci, 2022, doi:10.3390/brainsci12010110_

Round 1

Reviewer 1 Report

  • it would be appropriate to replace the term giddiness with the term dizziness
  • Line 15: tonal audiometry? PTA? it is necessary to specify better what is meant by normal
  • Line 16: "not severely affected" It is necessary to improve this definition.
  • Line 17: "calorics" caloric vestibular test
  • Line 31-32 (2019 Consensus document of the Committee for the International Classification 31 of Vestibular Disorders). Bibliographic citation is sufficient.
  • Line 40 "tortional" torsional 
  • Line 51-52 Recovery nystagmus is not frequently registered for two reasons: A) It does not take place in every individual with acute vestibular deafferentation (AVD). Bibliographic citation is necessary.
  • Line 65 velosity
  • Line 68 Only in Dix-Hallpike manuever? are you sure? Right Dix-Hallpike should be enough to describe the manouvre (posterior position is not necessary)
  • Line 71 Romberg and Unterberger tests were not severely affected. It is necessary to improve this definition.
  • Line 72-74 what is the meaning of the entire phrase?
  • Line 73 Celcius…
  • Line 74 dimenydrinate. Dosage?
  • Line 80 Calorics
  • Line 83 Cawthorn-Cooksey exercises. Bibliographic citation is necessary.
  • Line 86-87 Clinical and laboratory (caloric testing) examination was normal. Explain better the meaning of this statement and improve your English kindly.
  • Line 98-100 Vestibular neuritis is a disorder characterized by acute, isolated (without symptoms  from the anterior labyrinth) spontaneous vertigo due to unilateral vestibular deafferentation. The definition needs to be improved acute vestibular syndrome (AVS) with vertigo, nystagmus, gait disturbance and autonomic symptoms (nausea and vomiting), all of which must by convention last more than 24 h (Hotson and Baloh, 1998; Kattah et al., 2009).
  • Line 101 end organ. I advise you to rephrase this sentence better, talking about end organ and quoting the article by Yacovino you should also deal with the inconstant utricle and saccule involvement.
  • Line 137-139 Is this your impression supported by literature data? Argue your conviction better.
  • Line 166-170 It would be advisable to make the sentence discursive by avoiding a) and b).
  • Figure 3 needs modification to make it easier to understand. I suggest eliminating A, B, C, D to indicate the paths, perhaps replacing them with MVN and VN. I recommend representing MVN in Meniere and AVD in one figure, VN in Meniere and AVD in another figure.

Reviewer 2 Report

The article does not bring anything new. It is a presentation of a case that is well known and already elaborated in the medical literature (Kim, Hyun Ji et al. “Vestibular Neuritis With Minimal Canal Paresis: Characteristics and Clinical Implication.” Clinical and experimental otorhinolaryngology vol. 10,2 (2017): 148-152. doi:10.21053/ceo.2016.00948).

The article could be interesting only in the educational sense because the material is thoroughly studied, analyzed and presented. However, it is not suitable in this sense either, because the author uses outdated terminology and does not explain to the reader crucial things in acute vestibular syndrome (AVS) differential diagnosis, which are HINTS + tests.

Skew deviation test (SDT) was absent during the clinical examination, as well as finger rub and headshake tests which is unacceptable.

The caloric test was done by air so the result for CP cannot be taken as relevant.

Only cVEMP was done, but not oVEMP!

Is it quite a lot strange that video HIT is negative in the acute phase of the disease?

All in all, an interesting observation of the central recovery nystagmus, but Case report must be interesting, exceptional in all respects, special, not ordinary case.

Round 2

Reviewer 1 Report

No comments for authors.

Author Response

There were no comments for authors.

Thank you again for your useful suggestions and remarks.